# Combinations of Photodynamic Therapy with Other Minimally Invasive Therapeutic Technologies against Cancer and Microbial Infections

**DOI:** 10.3390/ijms241310875

**Published:** 2023-06-29

**Authors:** Sandile Phinda Songca

**Affiliations:** School of Chemistry and Physics, College of Agriculture Engineering and Science, Pietermaritzburg Campus, University of KwaZulu-Natal, Pietermaritzburg 3209, South Africa; songcas@ukzn.ac.za

**Keywords:** photodynamic therapy, sonodynamic therapy, photothermal hyperthermia, magnetic hyperthermia, anticancer, antimicrobial, combinations, nanomaterials chemotherapy, CAP, immunotherapy, radiotherapy

## Abstract

The rapid rise in research and development following the discovery of photodynamic therapy to establish novel photosensitizers and overcome the limitations of the technology soon after its clinical translation has given rise to a few significant milestones. These include several novel generations of photosensitizers, the widening of the scope of applications, leveraging of the offerings of nanotechnology for greater efficacy, selectivity for the disease over host tissue and cells, the advent of combination therapies with other similarly minimally invasive therapeutic technologies, the use of stimulus-responsive delivery and disease targeting, and greater penetration depth of the activation energy. Brought together, all these milestones have contributed to the significant enhancement of what is still arguably a novel technology. Yet the major applications of photodynamic therapy still remain firmly located in neoplasms, from where most of the new innovations appear to launch to other areas, such as microbial, fungal, viral, acne, wet age-related macular degeneration, atherosclerosis, psoriasis, environmental sanitization, pest control, and dermatology. Three main value propositions of combinations of photodynamic therapy include the synergistic and additive enhancement of efficacy, the relatively low emergence of resistance and its rapid development as a targeted and high-precision therapy. Combinations with established methods such as chemotherapy and radiotherapy and demonstrated applications in mop-up surgery promise to enhance these top three clinical tools. From published in vitro and preclinical studies, clinical trials and applications, and postclinical case studies, seven combinations with photodynamic therapy have become prominent research interests because they are potentially easily applied, showing enhanced efficacy, and are rapidly translating to the clinic. These include combinations with chemotherapy, photothermal therapy, magnetic hyperthermia, cold plasma therapy, sonodynamic therapy, immunotherapy, and radiotherapy. Photochemical internalization is a critical mechanism for some combinations.

## 1. Introduction

Several compounds known as photosensitizers absorb light energy and transfer it to oxygen in the triplet ground state to produce reactive oxygen in the singlet excited state. Following this, a series of other reactive oxygen species are produced in biological media. Upon excitation with light energy, most of these compounds can also react directly with the biological cell membrane and cytoplasmic components, including monosaccharides, nucleic and amino acid components of DNA, RNA, glycan, and protein molecules. Photodynamic therapy (PDT) is based on the initial absorption of light by the photosensitizer (PS) and the subsequent direct and oxygen-mediated reactions with the cell membrane and cytoplasmic cell components. These reactions initiate cell death by apoptosis or their cell destruction by necrosis. Thus, PDT requires the accumulation of the PS through targeted delivery to the disease site and cells, which either does not occur in normal healthy cells or does so to a relatively lower degree compared to the disease site and cells. The foregoing mechanism of PDT may be illustrated using a Jablonski diagram, which is shown in Figure 1.

Initially, when it was shown that the organic dye PSs of the porphyrin and phthalocyanine type accumulate preferentially in tumor tissue compared to normal host tissue, PDT was described as an anticancer therapeutic technology [1]. However, as the technology progressed through its current clinical translation, the need to improve this selective accumulation was pursued [2]. This triggered an avalanche of research aimed at enhancing the selective delivery of PSs to disease cells and sites compared to normal ones, which is still the main preoccupation of a large population of PDT researchers today [3]. Given that the predominant type II mechanism of PDT requires oxygen to be present in sufficient quantities in the disease microenvironment, it was soon realized that the well-documented hypoxia observed in such disease microenvironments as cancerous tumors and bacterial and fungal infections is another major limitation of PDT. This limitation also triggered more research to discover ways to improve oxygenation of the disease microenvironment to improve the efficacious application of PDT [4].

Another limitation of PDT is the depth to which light can penetrate human tissue to reach the PS and initiate the PDT reaction, which is no more than 2–5 mm [5]. While most skin tissue and other low-lying diseases are currently the best candidates for effective treatment, diseases that are located deeper than 5 mm or those that are located in dark tissues such as the liver, spleen, pancreas, and bones cannot be treated using PDT without using optic fiber light delivery, even when using light of frequency in the therapeutic window of the near-infrared region. Some of the deeper-lying diseases can be reached through optic fiber light delivery for activating the PS [6,7]. However, this approach may have infrastructure requirements that make it technology intensive and inaccessible to poor patients [8]. Sonodynamic therapy (SDT) is a recent iteration of PDT that was developed to overcome the light penetration depth challenge in PDT [9]. Instead of light, the SDT sensitizer, which is known as the sonosensitizer, is activated by ultrasound, which can go through the entire body, thus, reaching through dark tissues and bones [9]. For this reason, much research is dedicated to the application of SDT in periodontal, bone disease, and diseases of dark tissues such as the liver, spleen, and pancreas. X-ray-induced PDT (XPDT or PDTX) offers another way to overcome the penetration depth challenge of normal visible light-activated PDT. In this approach, the PS is activated by nanomaterial agents that are excited by X-rays to emit visible light, which excites the PSs [10]. Magnetic hyperthermia therapy (MGH) is currently in clinical use to treat glioblastoma multiforme located deep in the brain. MGH overcomes the tissue penetration depth challenge because the high-frequency alternating magnetic field radiates through brain tissue and the skull to elevate the temperature of glioblastoma multiforme tumors in the brain where magnetic nanoparticles are embedded for the purpose [11]. This review discusses a few combinations of PDT and MGH and problematizes the paucity of antimicrobial research applications. 

In most clinical PDT applications, the PS is still administered directly to the patient to reach the diseased tissue after systemic circulation [12]. Nowadays, however, much of the research aimed at enhancing the efficacy and selectivity of PDT uses nanomaterials as vehicles for carrying and delivering the PS to the diseased tissue for precision-targeted delivery and selectivity for the disease over host tissue [13]. Due to their small size and, therefore, high surface area to volume ratio, nanomaterials can adsorb, carry, and deliver large quantities of PSs. Like porphyrin and phthalocyanine organic dye PSs, evidence exists to show that the enhanced permeability and retention effect may drive the accumulation of nanomaterials in tumor tissue [14]. Therefore, nanoparticles have been widely researched for their potential to enhance the efficacy of PDT through enhanced targeted delivery of the PS to the diseased tissue [13]. In addition, nanoparticles have been identified that act as PSs by photosensitizing the production of reactive oxygen species [15]. While these nanomaterials render it unnecessary to load them with organic dye PSs for delivery to the disease site, when loaded with such PSs, the resulting nanoconjugates have been shown to have potent reactive oxygen-generating capabilities and PDT efficacy [15]. 

Nanomaterials have been used as a basis for combination therapies involving PDT [16]. This is generally achieved by synthesizing nanoconjugates that possess the characteristics required for PDT and those of the technology that is being combined with PDT in the same nanoconjugate [17,18]. For example, for combining PDT with MGH, magnetic nanoparticles are loaded with the PS [19]. This produces a nanoconjugate that responds to a high-frequency alternating magnetic field by generating heat, thus elevating the temperature beyond the tolerable limit of the disease cells in the microenvironment. Similarly, to combine PDT with photothermal hyperthermia therapy (PTT), photothermal nanoparticles are loaded with the PS, generating a nanoconjugate that responds to light irradiation by elevating the temperature beyond the tolerable limit of the disease cells in the microenvironment [20]. However, to combine PDT with chemotherapy, nanoparticles are loaded with a suitable chemotherapy drug as well as the PS to give a nanoconjugate that delivers the PS and the chemotherapy drug to the disease site and enables simultaneous PDT and chemotherapy [21]. This paper also discusses the combination of PDT with cold atmospheric pressure plasma (CAP) therapy. Although CAP does not necessarily require nanomaterials, in combination with PDT they act as a carrier and delivery agents for the PS [22].

## 2. Purpose Statement

This review will navigate some of the innovations that have emerged among the seven combinations with PDT to highlight the innovative applications of nanomaterials as the basis for combination drug design and obviate novel areas of further research. The review will focus on basic studies including the development of the nanoconjugates and their in vitro studies. However, mention is made of preclinical, clinical trials and clinical case studies where the discussion necessitates such. The review discusses several nanomaterial-based PSs and the impact of the nanomaterials on cell targeting, selectivity and stimulus-responsive PS and drug release in the cells as part of the overall discussion of efficacy.

## 3. Anticancer Photodynamic Therapy

When applied to cancer, PDT can act through a combination of cancer cell death and inhibiting the growth of tumors due to cancer cell growth inhibition. It relies on the preferential accumulation of the PS in cancer tissue, where it photosensitizes the generation of reactive oxygen species which kill cancer cells in the microenvironment of the disease. The origin of the clinical approach is attributed to the direct administration and some level of preferential accumulation of hematoporphyrin and hematoporphyrin derivative following systemic circulation [23]. Current focus is now paid to targeted nanomaterial-mediated PS administration. However, some researchers still administer the free PSs directly without nanomaterials. The integration of coumarin into amphoteric nanocapsules is an example of the administration of the PS encapsulated in organic nanomaterials with the aim of optimal systemic navigation due to the amphiphilicity of the nanomaterial and disease accumulation due to hydrophobicity of the PS [24]. 

Comparison of the in vitro efficacy of novel zinc (II) phthalocyanine-quinoline conjugate with Photofrin, on the other hand, illustrates the direct approach without incorporating the PS in any form of nanomaterial [25]. In this study, the zinc (II) phthalocyanine-quinoline conjugate and the Photofrin were administered to the cell lines as solutions in N, N-dimethylacetamide containing 3.8% polyoxyethylene-35-castor oil, and phosphate-buffered saline, respectively. The advantages of using nanoparticles as carrier and delivery agents for the PS, compared to using the free PSs, which include the improvement of solubility, biodistribution, uptake, retention, targeted cancer cell delivery, the reduction of self-degradation, and applications in combination therapies involving PDT, have been elucidated with numerous examples of metal nanoparticles encapsulated in various shells containing the PS [26]. In this regard, silver and gold nanoparticles are among the most widely studied [27]. While metal oxide nanoparticles are widely studied as carriers and delivery nanosystems of the organic dye type of PSs for targeted anticancer PDT [28], iron oxide nanoparticles have been highly prized for magnetic hyperthermia, magnetic tumor targeting and magnetic resonance imaging-assisted anticancer PDT [29,30,31,32] 

Cancer cell-targeted, controlled, and stimulus-responsive release of the PS is seen as the ultimate gold prize of nanomaterial-mediated PDT, through which deep metastatic cancer cells can be isolated for destruction before they form tumors [33,34]. Much research has therefore focused on metastatic cancer cell targeting for PDT. Photoimmunotherapy, for example, is a technology that combines PDT and immunotherapy, in which the PS is conjugated to a monoclonal antibody or biologically recognizable fragment thereof to target antigens actively and specifically for high-precision targeted drug delivery and induction of anticancer immune response [35]. Additionally, a peptide that is easily cleaved by the cysteine protease Cathepsin B was used as a linker for anchoring the PS Verteporfin onto the surface of gold nanoparticles so that upon cancer cell internalization of the nanoconjugate the PS is released after cleavage of the peptide linker [36]. 

## 4. Antimicrobial Photodynamic Therapy

From its origins as an anticancer therapeutic technology, PDT has evolved. It is now widely used against a variety of ailments, the top among which is antimicrobial PDT as applied against bacterial, fungal, and viral infections, as well as applications in the sanitization of the environment and pest control [37]. Antimicrobial PDT against many viral infections, such as the recent COVID-19 viral pandemic [38], bacterial [39] and fungal [40] infections, and environmental sanitation [41], have been widely investigated. Examples of applications of antibacterial PDT include periodontal disease [42,43], superficial wounds [44], and burns [45]. It is considered to hold great promise for the developing world, where bacterial infections cause a large number of deaths, in a global context where the widespread use of antibiotics is fuelling the rapid emergence of bacterial resistance [46]. 

The formation of biofilms drives the development of resistance with some bacterial and fungal infections through a multistage process involving planktonic microbial quorum sensing [47], surface adhesion, colony formation and maturation, and biofilm formation, with microbial detachment from biofilms leading to attachment elsewhere, starting the process all over again as the biofilm grows [48,49]. Throughout its life cycle, the bacterial cells remain suspended in, and otherwise embedded in the biofilm, which is made up of an extracellular polymeric substance matrix consisting of polysaccharides, proteins, peptides, nucleic acids, and lipids, making it difficult to reach the suspended bacterial cells [50,51,52]. Antibacterial PDT has been shown to overcome the biofilm-mediated bacterial resistance strategy by breaking down the extracellular polymeric substance matrix, thereby providing access to the bacterial cells that are embedded in it [53,54].

Nanoconjugates incorporating the PS in their structure and composition constitute the most widely reported antibacterial PDT strategy compared to the application of free PSs [55]. The structure of these nanoconjugates is designed to ensure labile incorporation of the PS into the conjugate so that it retains the capability to generate reactive oxygen species [56]. Several nanoparticles act as antibacterial PDT PSs capable of biofilm degradation and bacterial cell destruction on their own. The most widely reported include nanoparticles of porphyrins [57] and phthalocyanines [58], metallic silver [59] and metallic gold [60], and copper sulphide [61], as well as oxides of nanographene [57], zinc [62], and iron [63]. For example, the three different morphologies of copper sulfide nanostructures, including microspheres, nanosheets, and nanoparticles, were found to exhibit different antibacterial PDT activities against *E. coli*, and this was largely attributed to their different concomitant photothermal conversion coefficients upon irradiation with normal sunlight [64]. Most of the magnetic metal chalcogenide nanomaterials exhibit concomitant photothermal and magnetothermal conversion capability and are therefore used as agents for combination therapies involving PTT, MGH, and PDT. For example, iron oxide nanoparticles are used in MGH, PTT, and PDT combinations [63]. Furthermore, studies have shown that nanographene oxide and copper sulfide nanoparticles are efficacious PTT and PDT agents [57,61]. Bacterial infection stimulus-responsive release of the PS derived from cleverly designed nanoconjugates for antibacterial PDT exploits any of the characteristics of the biofilm microenvironment, such as lower pH, insufficient oxygenation, and the altered concentration of enzymes and hydrogen peroxide compared to normal tissue microenvironments [65,66]. Table 1 lists the some of the current applications of PDT.

## 5. Other Applications of Photodynamic Therapy

As the pathogenic cause of acne, bacterial and fungal growth that occurs when the hair follicles become plugged with oil and dead skin cells in the skin, especially on the face, may be treated with antibacterial PDT [70]. Wet age-related macular degeneration occurs when abnormal blood vessels grow in the back of the eye and damage the macula [84]. PDT destroys the macula blood vessels, after which new growth gives rise to normal blood vessels [85]. An accumulation of plaque in the inner lining causes atherosclerosis as a thickening or hardening of the arteries [86]. Fast-growing skin cells cause an itchy scalp and flakes on the skin known as psoriasis [87]. The recent emergence of viral pandemics, such as the recent pandemic caused by COVD-19, has triggered an avalanche of interest in antiviral PDT [88]. PDT has been shown to destroy viruses, bacteria, and fungi [89]. For this reason, the technology has been tested for sanitization of the environment, such as work surfaces, where the reactive oxygen species produced can kill microorganisms. It has also been tested as a pesticide against insect larvae [80]. From the foregoing discussion, it is easy to see why the applications of PDT have gone beyond cancer and bacterial infections to include the treatment of acne [70,71], wet age-related macular degeneration [72,73], atherosclerosis [74,75], psoriasis [76,77], and antiviral treatments [83,90], including herpes [67,68] and the COVID-19 [69] virus. Investigations have been conducted for environmental sanitization [78] and pest control [80,81] and why it is among the most commonly used therapeutic techniques in dermatology [82,91].

## 6. The Value Proposition of Combinations of Photodynamic Therapy

PDT has been described as a minimally invasive technology for treating an increasing range of ailments [92]. Several limitations of PDT have been cited, including the poor solubility of the PSs, low depth of the tissue penetration by the light used to activate the PS, the rather low oxygenation levels in most of the microenvironments of the candidate diseases against which it is used, sub-optimal levels of the PS that reach the disease after systemic circulation, and the poor selectivity for the disease at tissue and cellular levels [93]. These challenges limit the efficacy in vivo of PDT even for candidate PSs that show potent efficacy in vitro. Combinations of PDT with a range of other minimally invasive therapies is one way that has been explored to enhance the efficacy in any of the identified areas where its efficacy is limited [94]. The primary purpose of these combination therapies is to enhance efficacy and therefore promote clinical translation of the combination. 

Because the term combination has been used widely in drug development research, it is defined more precisely in this review to mean the combination of several technologies, in which each has recognized therapeutic applications. The non-invasive and minimally invasive technologies considered in this review have different physical and biological mechanisms, targeting different intracellular organelles and processes and eliciting several biological responses. Therefore, the composite mechanisms of the combinations merge to induce complex processes that can either augment, act independently, or act against each other. Because of the foregoing reasons, combination therapies have been evaluated on the basis of three different impacts on efficacy. When the efficacy of the combination therapy exceeds the sum of their individual efficacies, the outcome is synergistic. When the two have equal efficacy, the outcome is additive. When the combination efficacy is less than that of the sum of the individual efficacies, the outcome might be antagonistic [95]. For example, when the efficacy of the combination of PDT using meso-tetrakis(3-hydroxyphenyl)chlorin as the PS, and the platinum drugs carboplatin, cisplatin, and oxaliplatin, was evaluated in vitro using the methyl thiazole tetrazolium essay, synergism was observed with oxaliplatin and three different cancer cell lines. In contrast, additivity and antagonism were observed with the other drugs as shown in Table 2 [96]. One of the mechanisms of drug resistance is the prevention of drug entry into target cells [97]. A nanoconjugate consisting of a core of doxorubicin-loaded calcium carbonate and a shell of human serum albumin conjugated with the PS Rose Bengal was prepared so as to improve the endocytosis efficiency, thus overcoming the doxorubicin resistance. The enhancement of the antiproliferation effect in vitro was synergistic [98]. On the contrary, it was suggested more than two decades ago that the combination of radiotherapy with PDT is generally additive [99]. Much of the subsequent evidence has supported this observation [100]. An example of a subadditive outcome was reported from the evaluation of the efficacy of the combination of PDT using the PS methylene blue with chemotherapy using the drug erlotinib against the human epidermoid carcinoma A431 cell lines in vitro [101].

## 7. Applications of Nanomaterials in Combination Therapies with Photodynamic Therapy

Before the rise of nanotechnology into its current state of prominence, therapeutic studies on the applications of PDT were conducted by directly applying the free form of the PS. After a period of time during which the free PS was in the systemic circulation, preferential accumulation generally leads to enhanced concentration in cancerous tissue. The preferential accumulation of the PS in disease tissue and the subsequent photosensitization reaction, which produces reactive oxygen species, is the essential basis of PDT because it means that there is a higher concentration of the PS at the disease site and in the disease cells, compared to cells in other sites of the body after the systemic circulation period. However, the degree of this preferential accumulation of the free form of the organic dye PSs soon proved to be insufficient for the required levels of disease targeting and selectivity for disease cells over host tissue cells as the technology translated to the clinic. The direct approach to combination therapies involving PDT involved the independent administration or direct combination of the free PS and the agent of the therapy that was being combined with PDT without the aid of nanomaterials. Examples abound [102,103].

In addition to the common applications for enhancing the selectivity of the PS for the disease over host tissue cells in PDT, nanotechnology is widely used to facilitate various combinations of PDT with other noninvasive therapeutic technologies, resulting in further enhancement of efficacy [13]. Consequently, several innovations for target cell specificity started to emerge based on the elaborate design and fabrication of nanoconjugates that incorporate the PS. In any case, PTT and MGH are technologies commonly combined with PDT, which are already based on the use of nanomaterials. Another development of the application of nanotechnology in PDT was the expansion to applications on disease types other than cancers, including bacterial, fungal, and viral infections. The nanomaterial-mediated enhancement of target specificity in PDT, which implies reduced systemic dose requirements for copious PS delivery at the disease site and in the disease cells, shows great promise in research studies aimed at reducing the emergence of resistance, especially by microbial pathogens [98]. Nanomaterial-mediated selectivity for disease cells over host tissue also facilitated various chemotherapy combinations with PDT. One spin off from this is the possibility of repurposing chemotherapy drugs that had been rendered obsolete by the development of resistance against them [98]. 

Although the innovations differ widely in design and fabrication, the two main divisions of nanomaterials used for PSs and other drug delivery are the organic and inorganic nanoparticles [104]. In alignment with this classification, nanoparticles of organic compounds and those of entirely hydrocarbon materials are grouped under organic. In contrast, nanoparticles of metallic and metal chalcogenide materials are grouped under inorganic nanoparticles, as illustrated in Figure 2 [105]. Organic nanoparticles include all the nanomaterials fabricated from carbon compounds, whereas inorganic nanoparticles include all the nanomaterials fabricated from noncarbon-based compounds [106]. In reality, however, there is a continuum between organic and inorganic nanoparticles because some researchers classify nanoparticles of hydrocarbon materials such as graphene, fullerenes, and carbon nanotubes as inorganic instead of organic [107,108]. The incorporation of PSs into the nanoparticles can be accomplished with surface adsorption and absorption, covalent linking, and encapsulation. The incorporation of the PS into the nanoparticle can lead to the enhancement of the preferential accumulation because they are preferentially taken up and retained by disease cells due to the enhanced permeability and retention effect. In addition, the improvement of the stability and biocompatibility of the nanoconjugates in which the PS and other drugs are incorporated is used to ensure that during systemic circulation, nanoconjugates do not induce an immune response. 

## 8. Combinations of Photodynamic Therapy with Chemotherapy 

The anticancer and antimicrobial therapeutic approach that uses chemical compounds to kill fast-growing cells in the body is generally referred to as chemotherapy [109]. Chemotherapy drugs now in clinical use are mainly derived from basic research involving the prospecting of natural sources, followed by a series of standard chemical and biochemical research studies, including chemical synthesis and modification to facilitate subsequent structure-activity studies and the enhancement of efficacy and selectivity [110]. The candidate drug compounds translate to clinical applications and clinical case studies through preclinical studies and clinical trials. The rising incidence of resistance fueled PDT research [109,111] among other alternate approaches, most of which are showing promising results in clinical trials [112]. Chemical modification of known effective drugs is one of the methods used to overcome chemotherapy drug resistance. Recently, however, the combination of drugs used for chemotherapy with PDT has emerged, offering great promise for overcoming drug resistance, enhancing efficacy, and improving selectivity [111]. The chemotherapy approach generally includes antibiotic and anticancer chemotherapy, with antibiotic drugs used for antibiotic chemotherapy and anticancer drugs used in anticancer chemotherapy. Both have suffered different degrees of the development of resistance [109]. 

Combinations of antibiotic chemotherapy with PDT have generally shown improvement in the efficacy against several microbial pathogens [113]. This may be illustrated with the antimicrobial PDT study that was conducted on third-degree burn wounds in vivo using the PS protoporphyrin IX, combined with antibiotic chemotherapy using the broad-spectrum antibiotic drug ceftriaxone sodium. In this study, copious amounts of reactive oxygen species were produced upon photoactivation and synergistic inhibition of the growth of methicillin-resistant *S. aureus*, *E. coli*, and *P. aeruginosa* [44]. Moreover, the impact of PDT using the PS Chlorin-e6, in combination with antimicrobial therapy using the antibiotic drug ciprofloxacin, improves the inhibition of bacterial growth and therefore offers a promising antibacterial approach against urinary tract infections [114]. Optimization studies on antibacterial PDT in combination with chemotherapy, using the antibiotics ciprofloxacin, amikacin, and colistin (Figure 3), found that the best conditions for optimal bactericidal activity of the combination therapy were 100 μg/mL of the PS Chlorin-e6 and a light energy dose of 120 J/cm^2^ for the quinolone antibiotic ciprofloxacin against *Escherichia coli* suspensions [115].

The synergistic effects of PDT and antibiotic chemotherapy using the PS methylene blue and the antibiotic drug ciprofloxacin were observed with a PS concentration-independent and light dose-dependent biofilm reduction of 10–5.4 and 10–7 for *S. aureus* and *E. coli*, respectively [116]. Recently, the combination of antibiotic chemotherapy using the antibiotic drug ciprofloxacin **1** with antibacterial PDT using positively charged porphyrin **4** and phthalocyanine PS complexes **5** (Figure 4) resulted in *S. aureus* and *E. coli* biofilm log10 reductions of 7.05 and 7.20, respectively, at the PS-to-antibiotic concentration of 8 μM to 2 μg/mL and 8 μM to 4 μg/mL, respectively [117]. Several reviews have concluded that the combination of PDT with chemotherapy have synergistic outcomes. For example, Ahmed El-Hussein et al. (2021) concluded that most chemotherapy drugs used in combination with PDT have synergistic outcomes against lung cancer [118]. On the other hand, the benefits of aPDT in combination with antibiotic chemotherapy were identified by the reviews by Pérez-Laguna et al. (2019), including reductions in the duration of treatment, antibiotic drug dosage toxicity, and the development of drug resistance [115,119]. 

## 9. Photochemical Internalization

Although originally developed for the photochemical delivery of macromolecular drugs into the cytosol, the technology known as photochemical internalization (PCI) is uniquely useful for the internalization of PSs into disease cells by incorporating them into endocytic liposome vesicles [120,121]. Once inside the cytosol of the disease cells, the release of the macromolecular drugs from the vesicles is induced by light activation, which causes photodynamic disintegration of the endocytosed vesicle membrane in the cytosol. This releases the contents of the aqueous core of the endocytosed vesicle and those incorporated within the membrane bilayer into the cytosol of the disease cell. The concentration of the PS incorporated in the membrane of the endocytosed vesicle for PCI applications is high enough to induce the photodynamic rapture of the vesicle but not high enough to cause PDT-induced cell death [122]. Studies on PCI-mediated delivery of chemotherapy drugs have been widely reported [123]. For example, PCI delivery of bleomycin was shown to enhance its efficacy [124]. Therefore, PCI-induced chemotherapy drug release may be considered an efficacious combination of PDT with chemotherapy [125]. For PDT applications, however, the liposomal formulation of the PS uses sufficiently high concentrations of the PS to rapture the endocytosed liposome and cause PS-induced PDT cell death. For example, the treatment of the wet form of macular degeneration uses a liposomal formulation of benzoporphyrin derivative, marketed as Visudyne^®^, in which the concentration of the PS is high enough for PS-induced macular cell ablation [126]. Another example of the liposomal formulation is that of meso-tetrakis(3-hydroxyphenyl)chlorin, which is marketed as Foscan^®^ for the palliative treatment of head and neck cancers [127]. In their recent review, Rak et al. (2023) cited the incorporation of phthalocyanines complexes of aluminium, silicon, and iron, marketed as Photosens, Phthalocyanine-4, and Phtalox, respectively, for anticancer and antibacterial applications in addition to other liposomal formulations [128]. Therefore, the fine line between PCI and liposomal PS formulation PDT may be the PS/lipid ratio, with a low ratio being just enough to open the PCI vesicle by photodynamic action without causing further PS-induced cell ablation. In contrast, the PDT ratio must be high enough to cause PDT as well. This aligns with the recent review that described PCI as a specialized PDT approach for facilitating combinations of PDT with for example chemotherapy and immunotherapy [129]. 

## 10. Combinations of Photodynamic Therapy with Photothermal Therapy

As a result of the resonance of their low-lying resonance band electrons, nanoparticles can transform absorbed light energy into heat [130,131]. This photothermal energy conversion is the fundamental basis for the photothermal hyperthermia therapeutic technology, with potential applications against cancer [132] and microbial infections [133]. The technology is sometimes combined with PDT, enhancing the efficacy, mostly synergistically [134,135]. When nanoconjugates are used as carriers and delivery agents, the photothermal energy conversion may be used as an external stimulus to trigger the release of the PDT PS and chemotherapy drug cargo from the nanoconjugate [136]. Of the multitude of nanomaterials that exhibit this plasmonic resonance and photothermal energy conversion phenomenon, gold nanoparticles are among the most widely studied, and gold nanorods exhibit the highest photothermal conversion efficiency [137]. All nanorods show two plasmonic resonance absorption bands in their electronic absorption spectra, an intense band on the near-infrared side and a lower intensity on the blue side [138]. The near-infrared band is always the one used to activate the nanorods in photothermal conversion [139]. Some metal chalcogenide nanostructures, such as copper sulfide nanoparticles, exhibit both the photothermal conversion and generation of reactive oxygen species and, for this reason, act as excellent agents for the combination of PTT and PDT [140]. Because the challenge of tissue penetration depth of light is also a limitation of PTT as it is for PDT, it is therefore a limitation for the combination of PTT and PDT [141]. While some reviewers see PDT as a way to improve the sensitivity of the cancer microenvironment to PTT [114], others recognize the mutual efficacy enhancement between PTT and PDT in terms of the PTT-mediated heat generation enhancement of perfusion and oxygenation [135].

Therefore, loading nanoparticles that exhibit the photothermal energy conversion capability with PS molecules could potentially be used as a nanomaterial-mediated strategy for combining PTT and PDT [142]. Superparamagnetic iron oxide nanoparticles were coated with 3-aminopropyl silane and loaded with indocyanine green as the PS to give a nanoconjugate with a superparamagnetic iron oxide core, and a shell composed of 3-aminopropyl silane and labile indocyanine green. Under conditions of the combination of PTT and PDT, in vitro studies showed that this nanoconjugate completely eradicated Gram-negative *P. aeruginosa* and showed a 7-log reduction of Gram-negative *E. coli* [143]. However, it was ineffective against Gram-negative *K. pneumoniae* and Gram-positive *S. epidermis*. In this study, the conditions of the combination of PTT and PDT were confirmed by temperature elevation measurements and the generation of reactive oxygen species.

The self-assembly of a zinc (II) phthalocyanine functionalized with a 3-(dimethylamino) phenoxy group **6** to give a phthalocyanine nanoparticle is an example of nanomaterial constituted entirely from the organic dye PS, with plasmonic photothermal conversion capabilities. The reaction scheme is shown in Figure 5. As a result, it acts as a PS as well as a photothermal conversion agent, thus enabling the potent combination of PTT and PDT, completely inhibiting the growth of methicillin-resistant *S. aureus* at an ultralow nanomolar concentration under brief 655 nm laser irradiation [144]. This study confirms the photothermal conversion capability of self-assembled nanostructures formed by organic dye PS self-assembly as a general phenomenon. In another study, for example, the self-assembled indocyanine green PS nanostructure, which was coated with poly2-(dimethylamino)ethyl methacrylate in order to give a positively charged nanostructure, showed excellent photothermal and photodynamic antibacterial activity because it eradicated *P. gingivalis* in both in vitro and in vivo studies [145]. While several challenges of the combination were recently noted, the combination is hailed for reducing cytotoxicity, improving drug and PS solubility, and for the synergistic efficacy enhancement [142].

## 11. Combinations with Magnetic Hyperthermia

The application of a high-frequency alternating magnetic field on magnetic nanoparticles, known as magnetic hyperthermia, has the effect of increasing the temperature of the media in which the nanoparticles are embedded [146]. Like photothermal hyperthermia, it has gained ground as a therapeutic technology against cancer [147] and microbial infections [148]. It has been investigated for potential applications against atherosclerosis and thrombosis [149]. Unlike photothermal hyperthermia, however, magnetic hyperthermia does not have the light energy penetration depth challenge because the alternating magnetic field radiates through biological materials with relative ease for more than 500 mm [150]. The clinical translation of magnetic hyperthermia may be typified by the MagForce Charité Hospital in Berlin, where clinical trials of magnetic hyperthermia therapy on patients affected by glioblastoma multiforme, prostate, and pancreas tumors were performed [151]. It has been used in combination with PDT against cancer with promising results [152]. Like the combination of photothermal hyperthermia therapy with PDT, the mechanism of the combination of MGH with PDT involves the generation of heat and reactive oxygen species. In both combinations, reactive oxygen species are generated from PDT, whereas heat is generated from photothermal hyperthermia in the former combination and magnetic hyperthermia in the latter. The apparent lack of research studies on the application of magnetic hyperthermia in combination with PDT against microbial infection compared to its anticancer and other applications is quite curious [37]. It could be rationalized in terms of the complexity of the equipment and infrastructure requirements, given that the photothermal hyperthermia requirements are nearly trivial compared to the heavy investment requirements for magnetic hyperthermia equipment. 

Whereas a direct magnetic field has been used in combination with PDT using chlorin e6-laden mesoporous silica-capped iron oxide magnetic nanoparticles, without generating heat from an alternating magnetic field as in magnetic hyperthermia, it did push the magnetic nanoparticles to move deeper into the biofilm [153]. After capping with the curcumin as the PS, a nanoconjugate fabricated from superparamagnetic iron oxide nanoparticles showed good magneto-thermal conversion in application of a high-frequency alternating magnetic field alone. The nanoconjugate also showed excellent PDT upon irradiation with blue light alone without the application of a high-frequency alternating magnetic field, eradicating planktonic *S. aureus*. In spite of the indicative results from these studies, no experiments were conducted on the combination of magnetic hyperthermia with PDT using this nanoconjugate in this study using the simultaneous application of a high frequency alternating magnetic field and blue light. However, this research shows all the preparatory studies in this direction because these researchers have successfully capped superparamagnetic iron oxide nanoparticles with the PS curcumin and measured both the PDT efficacy and magnetothermal conversion curves separately, thus paving the way to the coveted joint application of magnetic hyperthermia and PDT against bacteria [154]. The paucity of reviews concerning the combination of PDT with MGH is punctuated by several research reports of nanomaterial-based innovations. One example is the hydrophilic core encapsulation of magnetic iron oxide nanoparticles in liposomes with the organic dye PS meso-tetrais(3-hydroxyphenyl)chlorin in the liposome membrane bilayer [155]. Another example is a Janus Nanobullet innovation from which release of the PS chlorine e6 from the mesoporous silica heads responds to pH and redox potential differencial on account of the disulfide linkage to the nanobullet heads [156].

## 12. Combinations of Photodynamic Therapy with Cold Atmospheric Pressure Plasma

When a gas is passed between electrodes with high direct or alternating voltage, a cold plasma discharge is generated with a high concentration of reactive gas species, including reactive oxygen species, the same as those generated by the photosensitization reaction of PDT [157]. The electric field removes some of the electrons from the gas species molecules, generating cations, radicals, and electrons, which, upon collision with the gas species, cause further ionization, radical formation, and release of electrons as the gas flows. All this takes place at ambient conditions of temperature and pressure, with minuscule temperature elevation, which is the makings of CAP, the constituents of which have been shown to be destructive to disease cells upon contact. The cold plasma discharge can be applied directly in wound healing in vivo and in the clinic [158], the same way it is applied during experimental studies onto the surface of the growth medium containing bacterial test cell lines in vitro [159]. CAP therapy has been combined with PDT in basic research studies [22] and in preclinical studies [160]. The rapidly increasing interest in this technology may be attributed to the fact that it generates reactive oxygen and reactive nitrogen species using plasma under ambient conditions at which the reactive gas species can be applied either directly against the disease sites or can be applied to appropriate stabilizing media and stored freezing at an ultra-low temperature for subsequent administration [161]. These are conditions for the combination of the CAP technology with other noninvasive technologies, such as PDT [162], chemotherapy [163], and electrochemotherapy [164], for antibacterial therapy in the healing of burns and wounds [165] followed by the stimulation of the regeneration of tissues [166], and in environmental antiseptic sanitization [167]. 

The CAP technology has been reported as offering great promise for anticancer applications [79,163]. Additionally, research studies have been increasing over the last two decades on the applications of the CAP technology against bacterial infection with excellent indications [168,169]. PDT and CAP therapy have been projected by the World Health Organization to become top among the novel technologies for eradicating bacterial pathogens in the backdrop of the alarming incidence of bacterial drug resistance [170]. New research studies have now emerged on the combination of PDT and CAP against various bacterial infections. For example, the adhesion and sealer penetration study using the pushout method showed a positive effect of the combination of the CAP technology with PDT. Still, no correlation was found between adhesion and sealer penetration [171]. 

The significance of infrastructure and devices in studying the combination of CAP and PDT against bacterial pathogens was highlighted with a recommendation for further study designs after a plasma device was found to be unsuitable in a comparative study of CAP and antimicrobial PDT [172]. A more recent study under different conditions has shown that CAP and PDT improved the pushout bond strength [173]. In a world-first and groundbreaking study, the combination of CAP and PDT against skin and wound pathogens has been reported to have synergistic enhancement of efficacy in vitro [174]. The combination of PDT with CAP has been scantily reviewed [175]. However, research reports of novel innovations that combine CAP with PDT have appeared in the literature. An example that was reported to enhance the efficacy against the human papillomavirus (HPV)-positive cervical cancer is a self-assembled polymer nanoconjugate incorporation of pheophorbide as the PS used in the combination with CAP [22]. In another example, PDT efficacy was enhanced when curcumin and methylene blue were used as the PSs in combination with CAP against *E. faecalis* from molar root canal infection [176]. It is therefore argued that because CAP is already in clinical use as plasma medicine [177], and PDT is also in clinical use with many FDA approved applications [178], the combination of PDT and CAP has high clinical feasibility and potential for rapid clinical translation.

## 13. Combinations of Photodynamic Therapy with Sonodynamic Therapy

Apart from radiation therapy which generates ionizing radiation, magnetic hyperthermia [146] and SDT [9] are two interventions that have emerged in recent research reports as significantly responding to the light penetration depth challenge of PDT. Magnetic hyperthermia converts some of the energy a high frequency alternating magnetic field to heat utilizing magnetic nanoparticles embedded in the target tissue [11,146]. SDT is somewhat similar to PDT in that low energy ultrasound radiation penetrating to a much higher depth compared to light is used [179]. SDT is based on ultrasonic activation of an SDT sensitizer to kill disease cells through the production of reactive oxygen species [180], generating oxidative stress and overwhelming the cellular redox homeostasis biochemical mechanisms. A combination of the two technologies, therefore, is conceptually quite easily achieved with the simultaneous application of light and ultrasound after the accumulation of the sonosensitizer at the disease site because many of the compounds that have been identified as sonosensitizers also act as PSs [181]. 

Several reviews on the investigations of the combination of PDT with SDT against cancer and bacterial pathogens have appeared in the literature [182,183]. Understandably, the combination is showing increasing interest on research into bone and dark tissue disease such as osteosarcoma [179], osteomyelitis [184], periodontal [185], pancreatic [186], hepatic [187], and brain [188] disease. Like PDT, the mechanism of SDT involves the generation of reactive oxygen species. Sonoporation plays a key role in the pharmacokinetics, uptake, and retention of the sonosensitizer [189]. The increasing number of patents and clinical case studies suggest that the translation to the clinic is underway. For example, a patent was filed by the Harbin Engineering University and Harbin Medical University in 2011 [190]. Prior to this, a patent was filed by the Science Group Pty. Ltd. [191]. 

Curcumin has been widely studied as an effective sonosensitizer against atherosclerosis [192,193] and bacterial [194] infection. A study of curcumin as a potential sono-PS for photo-SDT showed enhanced antibiofilm efficacy and healing of wounds infected with *A. baumannii* in mice [195]. This study illustrates the application of a single sensitizer in the combination of SDT with PDT. Rose Bengal [196] and Chlorin e6 [197] both act as PSs and sonosensitizers albeit to different degrees. The combination of sonodynamic therapy and PDT using both Rose Bengal and Chlorin e6 at molar ratios of 1:1 and 1:3 showed synergistic enhancement of the generation of reactive oxygen species and efficacy against methicillin-resistant *S. aureus* [198]. This study illustrates the application of dual sono-PSs in the combination of SDT with PDT, with each responding mainly to one excitation method. However, both are excitable by both stimuli. These studies illustrate the synergistic enhancement of antibacterial efficacy and the production of ROS. 

## 14. Combinations of Photodynamic Therapy with Immunotherapy

PDT [199] and SDT [200] induce host immunity. However, the immunity response is not always strong enough to inhibit bacterial and tumor growth and metastasis [201]. The antitumour immunity response is based on the binding of checkpoint proteins of cancer cells to proteins of the cell wall of T cells, thus preventing the T cells from attacking the cancer cells. Checkpoint blockade immunotherapy is based on preventing the cancer cell checkpoint proteins from binding to the proteins on T cells, thereby frustrating the induced immunity so that it fails to protect the cancer cells against the T cells [202]. The technology uses inhibitors, which attach either to the checkpoint proteins or their T cell targets, thereby blocking the immune response [203]. An illustration of how the checkpoint-blockade immunotherapy technology works is shown in Figure 6. 

The impact of combining SDT or PDT with checkpoint blockade immunotherapy is, therefore, to facilitate inate antitumor immunity and thereby improve efficacy. This is among the reasons that were put forward for this combination in reviews [94,204]. For example, the use of hematoporphyrin monomethyl ether as the sonosensitizer in combination with the immune modulator adjuvant R837, encapsulated in the same liposome, arrested the primary progression of Xenografts and prevented lung metastasis of breast and colorectal cancer cells [205]. In another example, the use of a cell membrane-cloaked nanometal-organic-framework sonosensitizer, combined with the R837 immune modulator adjuvant, showed similar synergistic efficacy enhancement [206]. 

Pang et al. (2019) reported the first antibacterial application of such technology using the application of cell membrane nanovesicles encapsulating the sonosensitizer with antibodies that neutralize the alphatoxin of methicillin-resistant *S. aureus* genetically engineered onto their outer surface [207]. Arguably, this research breaks new ground in the design of antibacterial immunotherapy in combination with SDT and other noninvasive therapeutic technologies. The authors refer to it as a bioinspired alternative approach, which aims to disarm as opposed to killing bacterial pathogens, in that it invokes the biology of bacterial immunity. Therefore, the combination of PDT, a localized therapeutic technology, with immunotherapy has the potential to be developed for the treatment of systemic bacterial infection by the induction of strong systemic immunity [208]. 

## 15. Combinations of Photodynamic Therapy with Radiotherapy

Together with chemotherapy and surgery, ionizing radiation has been among the topmost tools used against cancer in the clinic for a long time [100,209]. Possibly because both technologies exert their individual effects by generating radicals and, therefore, by inducing oxidative stress, PDT has been shown to enhance the effect of radiation therapy [210]. Unlike radiotherapy and SDT, which have higher penetration depths into human tissue, the light required to activate PDT PSs cannot penetrate normal tissue beyond a shallow depth of 2–5 mm [5]. Therefore, the combination of PDT or SDT with radiation therapy offers enough potential for researchers to conduct further investigations [211]. 

The research by Browning et al. (2021) showed that SDT reduced the tumor perfusion and vascularization of the combination of chemotherapy and radiation treatment and improved the Kaplan–Meier survival curves, using gemcitabine as the anticancer chemotherapy drug [212]. The research [100] and clinical studies [209] on the combination of PDT and radiotherapy have been reviewed. While these reviews have revealed that the combination has enhanced its efficacy against cancer, not much has been said about antibacterial applications. In any case, research is recommended since the reviews raise many questions such as light, ultrasound, radiation, and PS dosimetry, indicating that the successes are now overwhelmed by challenges.

## 16. Conclusions

To overcome each of the challenges of the direct application of PDT using free PSs, the technology has evolved to the currently predominant approach of conjugating the PS in nanomaterials and combining it with other minimally invasive therapeutic technologies. In this regard, examples of the research studies cited in this review are listed in Table 3. These examples show the current trends in the evolution of what, after nearly 50 years, is still regarded as a novel technology because it continues to translate to the clinic as it morphs in current research and clinical practice. The most widely studied applications are still anticancer and antibacterial. Due to the frequency and threat to the life of viral diseases in recent times, antiviral applications have appeared among the dominant antimicrobial applications as the scope expands widely from the original neoplastic and infectious diseases to inflammation, neovascular diseases, and environmental applications. 

Arguably, SDT has taken PDT and its combinations to greater depths. Similarly, greater depths are achieved from combinations with magnetic hyperthermia, where PDT affects mainly the superficial affliction and magnetic hyperthermia also affects the deeper-lying disease. Combinations of PDT with chemotherapy have the potential to revive the applications of chemotherapy drugs that have been relegated due to the development of drug resistance. In addition to enhancing PDT efficacy, combinations with photothermal and magnetic hyperthermia are also useful as thermal stimuli for the release of the PS from cleverly designed nanoconjugates. Combinations with immunotherapeutic technologies hold great promise to take the applications beyond localized to systemic anticancer and antibacterial treatments. The PCI technology enables cytosolic delivery of macromolecular drugs through the photodynamic reaction-mediated rapture of endocytosed carrier nenovesicles, and therefore uniquely enables PDT combinations, with for example chemotherapy and other minimally invasive technologies. In this combination, the photodynamic reaction is delicately used to rapture the carrier nanovesicle membrane to release the nanovesicle cargo into the cytosol without causing PDT-induced cell death. This review only presents examples of the current basic research in anticancer and antibacterial PDT together with combinations with chemotherapy, PTT, MGH, CAP, SDT, IT, and RT without being exhaustive. Although the combinations of PDT with each of these seven minimally invasive technologies offer alternatives to the current clinical tools, much research is still needed to evaluate their potential, elucidate their challenges, and obviate the development of clinical devices that will facilitate clinical translation and applications thereafter. 

Although much research has demonstrated the applicability of the combination of magnetic hyperthermia with PDT against cancer, there appears to be a notable paucity of similar research on the antibacterial applicability of this combination. This suggests a need to investigate this combination technology against bacterial pathogens in vitro initially followed by in vivo studies subject to the outcomes of the in vitro studies. The outcomes of the initial studies of the combination of PDT with CAP conducted by Aly F. (2021) [174] suggest that this combination could be a useful addition to the antibacterial PDT combinations. Hence, the groundbreaking research on the combination of CAP with PDT against common skin and wound pathogens promises to provide the much-needed proof of concept. Current studies and progress towards clinical translation on the combinations of PDT or deep tissue SDT with other minimally invasive technologies are at the cutting edge of the latest innovations that promise to be the latest anticancer and antimicrobial clinical tools. In addition, while PDT is essentially a localized therapeutic approach, its combination with immunotherapy has the potential to be developed to an antimicrobial therapeutic technology capable of treating systemic disease in the future. 

## Figures and Tables

**Figure 1 ijms-24-10875-f001:**
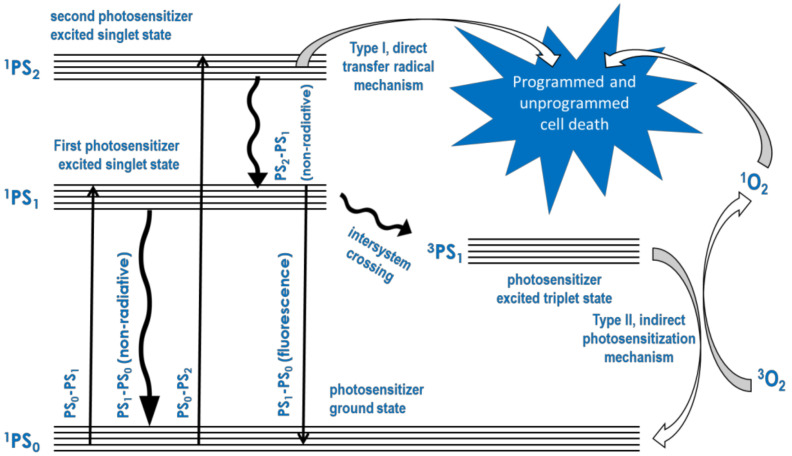
Mechanism of PDT shown by means of a Jablonski diagram.

**Figure 2 ijms-24-10875-f002:**
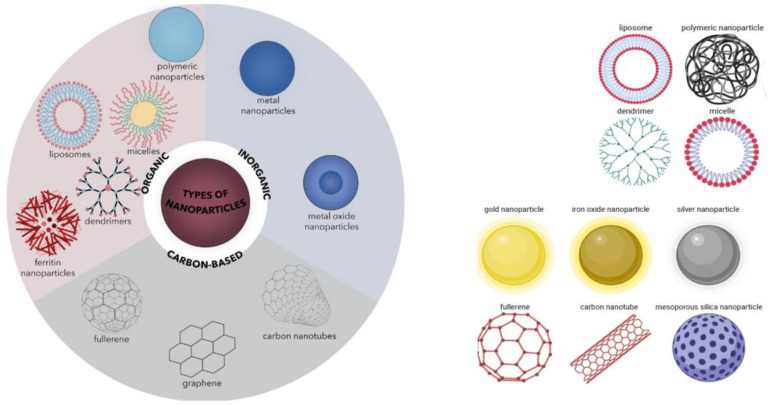
Classification of organic and inorganic nanoparticles. Reproduced from Spirescu et al. (2021) [107] and Greene et al. (2018) [108] under the creative commons attribution license 4.0.

**Figure 3 ijms-24-10875-f003:**
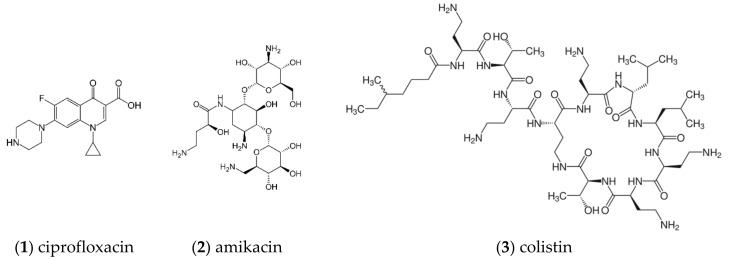
Chemical structures of the chemotherapy drugs ciprofloxacin, amikacin, and colistin.

**Figure 4 ijms-24-10875-f004:**
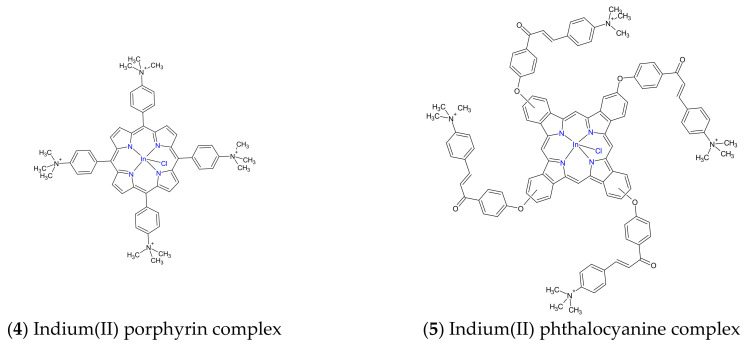
Chemical structures of the indium (II) porphyrin and phthalocyanine complexes.

**Figure 5 ijms-24-10875-f005:**
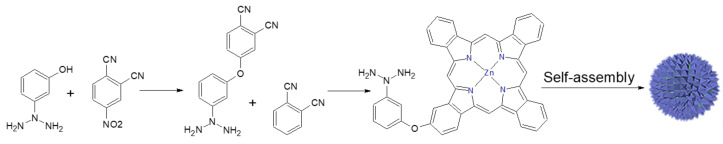
The synthesis and self-assembly of 3-(dimethylamino)phenoxy-functionalized zinc (II) phthalocyanine 6 to give a phthalocyanine nanoparticle. Adapted from Wang et al. (2022) [144].

**Figure 6 ijms-24-10875-f006:**
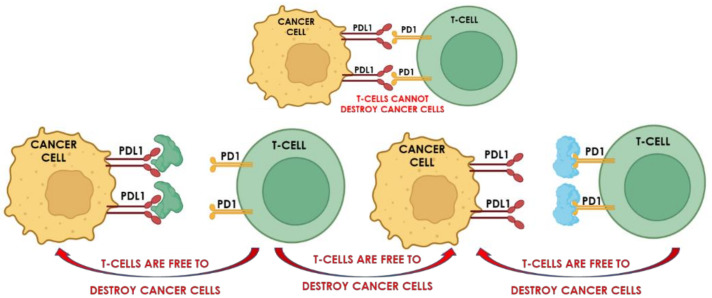
Schematic showing the principle of the anticancer immunotherapeutic technology known as checkpoint blockade.

**Table 1 ijms-24-10875-t001:** Some of the applications of photodynamic therapy.

	Applications and Description	References
1	Anticancer	[23,24,25,26,27,28,29,30,31,32,33,34,35,36]
2	Antimicrobial	[37,38,39,40,41,42,43,44,45,46,47,48,49,50,51,52,53,54,55,56,57,58,59,60,61,62,63,64,65,66]
3	Viral Herpes	[67,68]
4	SARS-CoV-2	[38,69]
5	Bacterial Acne	[70,71]
6	Wet age-related macular degeneration	[72,73]
7	Atherosclerosis	[74,75]
8	Psoriasis	[76,77]
9	Environmental sanitization	[37,78,79]
10	Pest control	[37,80,81]
11	Dermatology	[82,83]

**Table 2 ijms-24-10875-t002:** Synergistic, additive, and antagonistic outcomes of combinations of photodynamic therapy using meso-tetrakis(3-hydroxyphenyl)chlorin and chemotherapy against indicated cancer cell lines.

Combination of Photodynamic Therapy with Chemotherapy	The Outcome against the Indicated Cell Line
Photosensitizer	Platinum Drug	A-427	KYSE-70	SISO
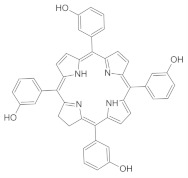	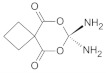	carboplatin	antagonistic	additive	additive
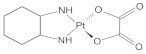	oxaliplatin	synergistic	synergistic	synergistic
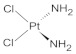	cisplatin	antagonistic	additive	additive

**Table 3 ijms-24-10875-t003:** Combination Therapies Involving Photodynamic Therapy.

	Free Photosensitizer	Example and Nanoconjugate Used		Reference
1	Anticancer	photofrin and zinc (II) phthalocyanine-quinoline conjugate		[25]
		photosensitizer-capped metal and metal chalcogenide nanoparticles		[26,27,28,29,30,31,32,33,34,35,36]
3	Antibacterial	porphyrins, phthalocyanines, metallic silver and gold, copper sulfide, and oxides of nanographene, zinc, and iron		[39,40,41,42,43,44,45]
	**Combinations**	**Example and nanoconjugate used**		**Reference**
4	chemotherapy	protoporphyrin IX/ceftriaxone sodium		[44]
		chlorin-e6/ciprofloxacin		[114,115]
		chlorin-e6/ciprofloxacin, amikacin, and colistin		[115]
		methylene blue/ciprofloxacin		[116]
		positively charged porphyrin 4, and phthalocyanine photosensitizer complexes and 5/ciprofloxacin		[117]
5	photochemical internalization	Photosensitizer	Chemotherapy Drug	
		meso-5,15-[(sulfonatophenyl)-10,20-(phenyl)]phorphyrin and meso-(tetraphenyl)porphyrin	saporin	[122]
		aluminum phthalocyanine disulfonate	bleomycin	[124]
6	photothermal therapy	superparamagnetic iron oxide nanoparticles/indocyanine green		[143]
		self-assembled zinc (II) phthalocyanine		[144]
		self-assembled indocyanine green coated with poly 2-(dimethylamino) ethyl methacrylate		[145]
7	magnetic hyperthermia	no research studies were found		
8	cold plasma therapy	protoporphyrin IX-loaded polymersome-mediated photodynamic therapy/cold atmospheric pressure plasma		[162]
		5-Aminolevulinic acid photodynamic therapy/cold atmospheric pressure plasma		[79]
		methylene blue photodynamic therapy/cold atmospheric pressure plasma		[171]
		chlorine p6 photodynamic therapy/cold atmospheric pressure plasma		[173]
		toluidine blue photodynamic therapy/cold atmospheric pressure plasma		[174]
9	sonodynamic therapy	curcumin photodynamic therapy/sonodynamic therapy		[192,193,194,195]
		Rose Bengal and Chlorin e6/sonodynamic therapy		[198]
10	immunotherapy	hematoporphyrin monomethyl ether/immune adjuvant R837		[205]
		membrane-cloaked nano-metal-organic-framework/R837 immune modulator		[206]
		sonosensitizer/cell membrane nanovesicles with antibodies that neutralize the alpha-toxin of MRSA genetically engineered onto their outer surface		[207]
11	radiotherapy	Rose Bengal-loaded oxygen microbubbles/radiotherapy		[212]

## Data Availability

Not applicable.

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
