# Peer review of "Combinations of Photodynamic Therapy with Other Minimally Invasive Therapeutic Technologies against Cancer and Microbial Infections"

_ijms, 2023, doi:10.3390/ijms241310875_

Round 1
Reviewer 1 Report
The manuscript is well written. It is very interesting and detailed about application of PDT with other methods. However the title should be approved. Also although the review comprises different methods there is a lack of one PDT method - Photochemical internalisation, method invented by Kristian Berg. And there are a lot of manuscripts describing the method. Also latest reviews in the field are not included. English should be approved, especially the small titles as should be better formulated : The value proposition of combinations of photodynamic therapy, Nanomaterials as a requirement for combination therapies .
All in all, it is a good and detailed review which should be helpful for readers.

English is readable, if it is approved will be better. Some titles are heavy and should be preformulated.
Author Response
Review responses.
|
|
Reviewer 1 |
Response |
|
1 |
The manuscript is well written. It is very interesting and detailed about the application of PDT with other methods. |
Thank you |
|
2 |
However, the title should be approved. |
Titles have been revised to be more clear and indicative of the ensuing discussion. |
|
3 |
Also, although the review comprises different methods, there is a lack of one PDT method Photochemical internalization, a method invented by Kristian Berg. And there are a lot of manuscripts describing the method. |
This has been done. |
|
4 |
Also, latest reviews in the field are not included. |
At least two reviews of each of the seven combinations with PDT have now been cited. The contribution of these reviews are listed on table 2. |
|
5 |
English should be approved, especially the small titles as should be better formulated: The value proposition of combinations of photodynamic therapy, Nanomaterials as a requirement for combination therapies. |
This has been done |
|
6 |
All in all, it is a good and detailed review which should be helpful for readers. |
Thank you |

Reviewer 2 Report
1. The use of English in the manuscript is relatively poor. I recommend that the manuscript undergo careful proofreading for language and readability. For example, these two sentences in the abstract are difficult to follow: “The core value proposition of combinations of photodynamic therapy is that they enhance the leading alternative and are proposed to overcome challenges of the leading clinical anticancer and antibacterial methods, mainly due to its inherent barriers against the emergence of resistance and its rapid development of targeted and high precision therapy. Combinations with chemotherapy and radiotherapy and demonstrated applications in mop-up surgery adventitiously promise to repurpose these top three clinical tools.” There are other language issues that crop up frequently throughout the manuscript. For example, line 457: “These are the conditions for the combination of CAP technology with other non-invasive technologies…” yet it’s not clear what exactly the author is referring to here. Line 477: “In a world-first and groundbreaking study…” seems like editorializing without any further explanation of what is meant.
2. The introduction is very long, and unfocused. There is only one reference, yet many of the statement made in this long introduction should be referenced. Furthermore, the introduction makes no mention of any of the 302 reviews of photodynamic therapy (PDT) and cancer that can be found currently on Medline. In the available literature, one can not only find many reviews of PDT and cancer, but also PDT combined with other therapies, such as with chemotherapy, sonodynamic therapy, etc. Consequently, the author does not clarify what is novel in the present review. I suggest the manuscript be focused clearly on the combinations of PDT. This focus would allow the introduction to be shortened considerably, would allow the author to refer to previous reviews, and would allow a clear statement of novelty.
3. Because the paper focuses on the (multiple) outcomes of cancer and microbial infections, the reader might suspect that the review is more about the technology rather than specific recommendations for treatment. It’s not clear, though. The manuscript ends up taking a rather long, winding, yet in many cases superficial, “tour” of various results related to various outcomes. Consequently, it’s not clear who is the targeted audience for this review. I believe that increased focus, with accounting for what information has been covered in previous reviews, would help in this regard.
The language suffers, at times rather dramatically, as suggested in comments to the author.
Author Response
Review responses.
|
1 |
1. The use of English in the manuscript is relatively poor. I recommend that the manuscript undergo careful proofreading for language and readability. For example, these two sentences in the abstract are difficult to follow: “The core value proposition of combinations of photodynamic therapy is that they enhance the leading alternative and are proposed to overcome challenges of the leading clinical anticancer and antibacterial methods, mainly due to its inherent barriers against the emergence of resistance and its rapid development of targeted and high precision therapy. Combinations with chemotherapy and radiotherapy and demonstrated applications in mop-up surgery adventitiously promise to repurpose these top three clinical tools.” |
These have been corrected. Please read lines 18-29. |
|
2 |
There are other language issues that crop up frequently throughout the manuscript. For example, line 457: “These are the conditions for the combination of CAP technology with other non-invasive technologies…” yet it’s not clear what exactly the author is referring to here. Line 477: “In a world-first and groundbreaking study…” seems like editorializing without any further explanation of what is meant. |
Reference is being made to the immediately preceding sentence, which describes “the ambient conditions at which the reactive gas species can be applied either directly against the disease sites or can be applied to appropriate stabilizing media and stored for subsequent administration.” These conditions enable the combination of the CAP technology with other minimally invasive technologies such as PDT. |
|
3 |
2. The introduction is very long and unfocused. There is only one reference, yet many of the statements made in this long introduction should be referenced. Furthermore, the introduction makes no mention of any of the 302 reviews of photodynamic therapy (PDT) and cancer that can be found currently on Medline. In the available literature, one can not only find many reviews of PDT and cancer but also PDT combined with other therapies, such as with chemotherapy, sonodynamic therapy, etc. Consequently, the author does not clarify what is novel in the present review. I suggest the manuscript be focused clearly on the combinations of PDT. This focus would allow the introduction to be shortened considerably, would allow the author to refer to previous reviews, and would allow a clear statement of novelty. |
The introduction needs to include a detailed discussion of the mechanism of PDT because this mechanism is used as a basis of the discussions of the combinations with the seven other minimally invasive technologies. It also needs to elaborate on the challenges and limitations of PDT, because the purpose of these combinations is to ameliorate or eliminate them. Furthermore, the introduction now mentions many reviews that are relevant in supporting the purpose of the review.
Therefore, the introduction has been revised to focus on combination therapies, the main purpose of the review. |
|
4 |
3. Because the paper focuses on the (multiple) outcomes of cancer and microbial infections, the reader might suspect that the review is more about the technology rather than specific recommendations for treatment. It’s not clear, though. The manuscript ends up taking a rather long, winding, yet in many cases superficial, “tour” of various results related to various outcomes. Consequently, it’s not clear who is the targeted audience for this review. I believe that increased focus, with accounting for what information has been covered in previous reviews, would help in this regard. |
The paper is indeed about technologies, specifically combination technologies. Hence recommendations are not for treatment but rather for more research in combination technology innovations. The abstract lists the key technologies in combination with PDT which are the focus of the review; chemotherapy, photothermal therapy, magnetic hyperthermia, cold plasma therapy, sono-dynamic therapy, immunotherapy, and radiotherapy. As a result, the subsections of the review discuss these combinations with PDT in turn, after a background discussion of mechanism in the introduction, anticancer PDT, aPDT, other applications of PDT, the value proposition of combinations of PDT, and the use of nanomaterials as a requirement for combination therapies. |
|
5 |
The language suffers, at times rather dramatically, as suggested in comments to the author. |
The areas where language is problematic in the manuscript have been revised and simplified for better readability and clarity. |

Reviewer 3 Report
This review discusses assorted uses of PDT and other procedures for treatment of malignant indications and microbial infections. While there is a potentially useful collection of recent information, it is not always made clear which results refer to strictly in vitro protocols and which relate to animal or clinical results. It is easy to eradicate malignant or microbial cell types in cell culture. When it comes to animal models or clinical studies, problems can arise. Systemic microbial infections cannot be treated with PDT. With regard to combinations involving PDT and ionizing radiation or chemotherapy, this might be feasible for well-localized tumors and for eradicating remaining malignant cells at sites of surgical debulking.
Going through the review, line 50 claims that clinical studies were ‘insufficient to be of . . . value’, but use of Photofrin did receive FDA approval in the US and elsewhere. Some efficacy was demonstrated for this approval. The term ‘disease cells’ (line 52) should be replaced by ‘malignant’ here and elsewhere.
The initial section of this review is essentially devoid of references. Ref. 2 does not occur until line 124. Metastatic cell targeting is mentioned in line 150, but the location of these cells would need to be known in order to direct light to the appropriate site(s). It may be feasible to photosensitize tissues to light by targeting procedures, but light is needed.
There is a discussion of antimicrobial PDT. The large number of deaths cited in line 168 will mainly involve systemic infections for which PDT will be of little value. Some of the indications noted in Table 1, e.g., sanitation and pest control, might be more readily dealt with using other approaches. What unique properties of PDT would suggest this would be better than chemical disinfectants, UV light, etc.?
Line 247 refers to an ‘essay’, which should be corrected to ‘assay’. This is the MTT assay which only measures the activity of mitochondrial dehydrogenases at a single time point, not true viability. In this section, the nature of the study is seldom revealed: cell culture, animals, clinic?
Considerable attention is given to MGH (magnetic hyperthermia). How useful is this procedure in anything but cell culture? It is feasible to provide alternating magnetic fields in a clinical situation? The term ‘disease cells’ (line 93) needs to be replaced by something more descriptive of the target. Use of PDT for treatment of macular degeneration was proposed in earlier days, but this appears to have been replaced by anti-VEGF therapy.
Mention is made of the potential for treatment of viral infections, but these tend to be systemic. PDT might have a limited level of efficacy for strictly local infections, but there are other procedures, e.g., treatment with cytosine arabinoside, that may be equally effective. Mention is made of CAP + PDT (Ref 107). On line 599, this is incorrectly cited as (2021). Ref 107 relates only to in vitro studies. How feasible is this approach for clinical applications? Examples are indicated in Ref. 107.
Generally adequate. Some terminology issues are raised in the review.
Author Response
Review responses.
|
1 |
This review discusses assorted uses of PDT and other procedures for the treatment of malignant indications and microbial infection signs. While there is a potentially useful collection of recent information, it is not always made clear which results refer to strictly in vitro protocols and which relate to animal or clinical results. It is easy to eradicate malignant or microbial cell types in cell culture. When it comes to animal models or clinical studies, problems can arise. |
Thanks for a comment that is central to the review. It should have been mentioned quite early in the review that because the aim is to promote therapeutic technologies, the focus is on the basic research of drug development through synthesis and nanotechnology incorporation and in vitro studies. Where necessary in-vivo studies, clinical trials and clinical case studies are mentioned. The reviewer is right, in vitro studies are easy to conduct and serve merely as an indicator on whether to proceed to pre-clinical studies. |
|
2 |
Systemic microbial infections cannot be treated with PDT. With regard to combinations involving PDT and ionizing radiation or chemotherapy, this might be feasible for well-localized tumors and for eradicating remaining malignant cells at sites of surgical debulking. |
The importance of this comment is that it highlights one of the limitations of PDT currently. This is mentioned among the challenges that have yet to be tackled in the future, as part of the conclusion. However, one combination that has potential to be developed for systemic applications is PDT and immunotherapy. This is mentioned in the conclusion in the revised manuscript. |
|
3 |
Going through the review, line 50 claims that clinical studies were ‘insufficient to be of . . . value’, but the use of Photofrin did receive FDA approval in the US and elsewhere. Some efficacy was demonstrated for this approval. The term ‘disease cells’ (line 52) should be replaced by ‘malignant’ here and elsewhere. |
Many first generation photosensitizers are currently approved for various applications in the clinic. Yet their preferential accumulation is still insufficient to be of competitive therapeutic advantage. As a result much research is ongoing to improve this aspect of preferential accumulation. The revised manuscript has replaced “therapeutic value” with “therapeutic advantage”. |
|
4 |
The initial section of this review is essentially devoid of references. Ref. 2 does not occur until line 124. Metastatic cell targeting is mentioned in line 150, but the location of these cells would need to be known in order to direct light to the appropriate site (s). It may be feasible to photosensitize tissues to light by targeting procedures, but light is needed. |
References are inserted in the initial sections in such a way that every contribution to the discussion is supported by a reference which gives an example or a full review of the contribution. As a result, more than 50 additional references are used to support the discussion. |
|
5 |
There is a discussion of antimicrobial PDT. The large number of deaths cited in line 168 will mainly involve systemic infections for which PDT will be of little value. Some of the indications noted in Table 1, e.g., sanitation and pest control, might be more readily dealt with using other approaches. What unique properties of PDT would suggest this would be better than chemical disinfectants, UV light, etc.? |
PDT and combinations thereof are viewed as having promise for the developing world where a large number of deaths occur due to bacterial infectious diseases because it can be used for localized disease as the reviewer points out. In addition it has many potential applications in environmental sanitation. Environmental contamination is one of the major contributors to the microbial related deaths in the developing world. |
|
6 |
Line 247 refers to an ‘essay’, which should be corrected to assay. This is the MTT assay which only measures the activity of mitochondrial dehydrogenases at a single time point, not true viability In this section, the nature of the study is seldom revealed cell culture, animals, clinic? |
“essay” has been replaced with “assay” in this line, thanks for the correction. The MTT is widely used and highly considered as a reliable standard in-vitro assay. This review focuses on in vitro studies, only mentioning in vivo and clinical studies where necessary. This is made clear in the purpose statement. |
|
7 |
Considerable attention is given to MGH (magnetic hyperthermia). How useful is this procedure in anything but cell culture? Is it feasible to provide alternating magnetic fields in a clinical situation? The term ‘disease cells’ (line 93) needs to be replaced by something more descriptive of the target. The use of PDT for the treatment of macular degeneration was proposed in earlier days, but this appears to have been replaced by anti-VEGF therapy. |
Magnetic hyperthermia is now in clinical translation by exactly utilizing the high frequency alternating magnetic field technology. The clinical translation of magnetic hyperthermia may be typified by the MagForce Charité Hospital in Berlin, where clinical trials of magnetic hyperthermia therapy on patients affected by glioblastoma multiforme, prostate and pancreas tumours were performed. This is true. Comparative research studies appear to have continued. I chose not to be distracted by the further developments in this area beyond mentioning the once highly successful PDT application using verteporfin. |
|
8 |
Mention is made of the potential for treatment of viral infections, but these tend to be systemic. PDT might have a limited level of efficacy for strictly local infections, but there are other procedures, e.g., treatment with cytosine arabinoside, that may be equally effective. Mention is made of CAP + PDT (Ref 107). |
Nevertheless, researchers are experimenting not only with PDT against systemic viral infection, but also combinations of PDT with other technologies. Once again, it serves little purpose in this review to go into this discussion beyond mentioning the emerging research on PDT and combinations against viral infections. |
|
9 |
On line 599, this is incorrectly cited as (2021). Ref 107 relates only to in vitro studies. How feasible is this approach for clinical applications? Examples are indicated in Ref. 107. |
The reference has been changed to Aly at al. (2019), thanks for the correction. Because CAP is in clinical use (plasma medicine) and PDT is also in clinical use (many FDA approved applications), the combination of PDT and CAP has high clinical feasibility. |
|
10 |
English is generally adequate. Some terminology issues are raised in the review. |
English usage has been completely revised. |

Round 2
Reviewer 2 Report
1. The abstract has some bizarre use of words and language. I suggest further proofreading.
2. The Introduction is still long and doesn’t tell the reader the purpose of the review. In my view, the reader deserves a brief, clear rationale for the review (what information is missing from the literature?) and a brief, clear outline of how the author plans to fill in the gaps.
In my opinion, the author should make clear:
What will be addressed in this review?
Why is this review of photodynamic therapy needed after hundreds of others have been published?
What is new here, and what is overlap with previous reviews?
3. The Introduction should also, in my opinion, how this review is different than reviews of the photodynamic therapy literature:
i) Photodynamic therapy – mechanisms, photosensitizers and combination (Kwaitkowski, 2018)
ii) Photodynamic therapy (Rkein, 2014)
iii) Photodynamic therapy of cancer: an update (Agostinis, 2011)
iv) Photodynamic therapy to control microbial biofilms (Warrier, 2021)
v) Cell death in photodynamic therapy: From oxidative stress to anti-tumor immunity (Donohoe 2019)
vi) Photodynamic therapy: A clinical consensus guide (Ozog, 2016)
vii) 1,679 other reviews of photodynamic therapy in Medline
In several places, the author's choice of words and phrases is still questionable. I'm not sure why the author is resisting a careful proofreading from a competent writer of English.
Author Response
Responses to reviewer 2 Round 2
|
# |
Comment |
Response |
|
1 |
The abstract has some bizarre use of words and language. I suggest further proofreading. |
Thanks, but the comment still leaves the author guessing which words are the “bizarre” words, and where is the “bizarre” language. I have read and re-read the abstract several times. I have not come across “bizarre” words, and “bizarre” language. Regrettably, therefore, it is not possible to effect remedial and corrective editing responding to this comment. |
|
2 |
The Introduction is still long and doesn’t tell the reader the purpose of the review. In my view, the reader deserves a brief, clear rationale for the review (what information is missing from the literature?) and a brief, clear outline of how the author plans to fill in the gaps.
In my opinion, the author should make clear:
What will be addressed in this review?
Why is this review of photodynamic therapy needed after hundreds of others have been published?
|
The introduction is necessarily long even after some reduction after the first review for three reasons: (1) It includes a discussion of mechanism of PDT in lines 3 to 50. “bizarre” words, and where is the “bizarre” language. (2) It also includes a historic background which introduces the limitations of PDT which have triggered the rising research on combination therapies, seven of which are the main focus of the review, in lines 53 to 89. (3) The review also introduces the use of nanomaterials because these have become the principal mechanistic tools upon which most combinations with PDT, including the seven, which are the focus of the review, are based. This is covered in lines 105 to 121.
There are several writing styles for reviews, and yes, the one recommended by the reviewer is a dominant style, especially for book chapters. In this style, the introduction builds up a series of challenges and ends with a statement of purpose, addressing the challenges. In this review, this is done in lines 24 to 32 of the abstract and it is very clear there what the purpose of the review is.
There is also a writing style in which the articulation of purpose is integrated with the discussion of the challenges. In this review this is done in lines 53 to 121, where, once again it is quite clear what are the challenges and what research the review presents as some of the solutions.
Yes, the reviewer presents their opinion of how the review should be structured. It is an honest and laudable opinion, but one which runs against the writing style of this review.
Furthermore, the reviewer appears to have missed the point that the review is not about PDT, but rather about seven specific combinations thereof. This is evident in the reviewer’s lamentations “Why is this review of photodynamic therapy needed after hundreds of others have been published?”, and “What is new here, and what is overlap with previous reviews?”. There are no reviews in the literature focusing on the seven technologies considered in this review. This review is also quite clear that these technologies are discussed because they are considered to be among the top in overcoming some of the challenges presented in the introduction (see lines 24 to 27: Due to their enhancement of efficacy observed during in-vitro and preclinical studies, clinical trials and applications, and clinical case studies, and their potentially facile applications, and rapid escalation of clinical translation, some of the combinations with photodynamic therapy have become prominent research interests.). |
|
3 |
The Introduction should also, in my opinion, how this review is different than reviews of the photodynamic therapy literature:
i) Photodynamic therapy – mechanisms, photosensitizers and combination (Kwaitkowski, 2018)
ii) Photodynamic therapy (Rkein, 2014)
iii) Photodynamic therapy of cancer: an update (Agostinis, 2011)
iv) Photodynamic therapy to control microbial biofilms (Warrier, 2021)
v) Cell death in photodynamic therapy: From oxidative stress to anti-tumor immunity (Donohoe 2019)
vi) Photodynamic therapy: A clinical consensus guide (Ozog, 2016)
vii) 1,679 other reviews of photodynamic therapy in Medline |
· While thanking the reviewer for this comment, I cannot stress enough how the reviewer has missed the point of the review as already explained.
· The reviewer is right, there are thousands of reviews on PDT. There are none that isolate the seven combinations presented in the current review to show what they offer from a basic research point of view.
· Some reviews have been identified which focus on one of the seven combinations. These have been valuable sources of information for the current review. For this reason, they are gracious cited in the current review.
· The author would gladly alter their standpoint with regard to this comment if a few reviews (even one) could be given which integrate all seven combinations to show the innovations used and their in vitro performance.
· The author has therefore resisted the temptation to write a review of the type recommended by the reviewer, because nothing new would be offered. |
|
4 |
Comments on the Quality of English Language In several places, the author's choice of words and phrases is still questionable. I'm not sure why the author is resisting a careful proofreading from a competent writer of English. |
The English usage in the review was severely softened and simplified to enhance readability and clarity. From an extensive record of reviewing manuscripts for IJMS, and other journals, I am confident that the language is not an issue anymore. The author is used extensively as a reviewer of English language usage. |

Round 3
Reviewer 2 Report
Regarding some examples of the use of English in the abstract – which is, unfortunately, reflective of the entire paper – for starters, the first sentence is awkward. Continuing, how many readers will know what is meant by “adventitiously promise” here? (Although I know what the words mean, I don’t honestly know what the author is talking about.) Another example is the following sentence: “Due to their enhancement of efficacy observed during in-vitro and preclinical studies, clinical trials and applications, and post-clinical case studies, and their potentially facile applications, and rapid escalation of clinical translation, some of the combinations with photodynamic therapy have become prominent research interests.” In addition to being one of many other awkward sentences, what is meant by “facile applications” here? It is unclear to me, and likely will be to most readers. The reviewer is not a copy editor and will not provide a long list. Suffice it to say that the writing here has some weaknesses that are not simply style-relate. I believe it would benefit from proof-reading, not only because I believe that reading the manuscript will be a “slog” for most careful readers, but in many cases the author’s meaning is not entirely clear.
Regarding the focus of the review, the author states: “Furthermore, the reviewer appears to have missed the point that the review is not about PDT, but rather about seven specific combinations thereof.” My point was that the focus is never entirely clear at the outset, and this is not unrelated to the issue raised in my previous comment. A clear, simple statement of purpose up front, before the reader goes through many paragraphs of complexly-worded introduction, would be helpful, I think.
Regarding the overall structure of the manuscript, including clarifying what is novel, what is well-established, etc., I agree with the author that it may not be optimal for certain style preferences, of which there are many. In any case, better use of English and clearer outline of the author’s purpose in writing the review should help.
Finally, I do not intend to offend the author and apologize if I have done so. I offer this advice for the sake of the author, the journal, and the readers of the journal. The author is to be commended for compiling and presenting a wide range of information.
I have tried to provide the author with some examples of how to possibly make the writing in the manuscript more enlightening and less frustrating for the reader. I personally believe that "moderate" editing is optimal, but I have selected "minor" editing to account for the differences in style noted by the author in the response letter. That is my suggestion, but it's entirely up to the author and Editor(s).
Author Response
Responses to reviewer 2 Round 3
|
# |
Comment |
Response |
|
1 |
Preamble |
I must start by apologizing to the reviewer. Clearly my second response must have come across as one offended author. Upon reading it again, yes, indeed, the reviewer’s judgement is correct. I was somewhat offended, unjustifiably so, and that is why I must open the response with an apology. Reviewers serve a fundamental purpose of quality assurance – to maintain standards and enhance not only the communication of science, but also the ranking of the high impact journals through which we benefit from such science. My sincere apology.
|
|
|
Regarding some examples of the use of English in the abstract – which is, unfortunately, reflective of the entire paper – for starters, the first sentence is awkward. Continuing, how many readers will know what is meant by “adventitiously promise” here? (Although I know what the words mean, I don’t honestly know what the author is talking about.) Another example is the following sentence: “Due to their enhancement of efficacy observed during in-vitro and preclinical studies, clinical trials and applications, and post-clinical case studies, and their potentially facile applications, and rapid escalation of clinical translation, some of the combinations with photodynamic therapy have become prominent research interests.” In addition to being one of many other awkward sentences, what is meant by “facile applications” here? It is unclear to me, and likely will be to most readers. The reviewer is not a copy editor and will not provide a long list. Suffice it to say that the writing here has some weaknesses that are not simply style-relate. I believe it would benefit from proof-reading, not only because I believe that reading the manuscript will be a “slog” for most careful readers, but in many cases the author’s meaning is not entirely clear.
|
Once again, the reviewer makes a strong point. Examples were requested, and examples were given. Obviously, there is more. There are just typifying examples. Therefore, there is no point of reducing the valuable work of the reviewer into that of an English editor. For this reason, I would like to avail the editing services of the journal to work on the entire manuscript until it reaches the standard of clarity and readability commensurate with IJMS.
I have made a gallant effort, however, to soften the two examples given here. Reading the second example for instance, one cannot help but notice an excessively long sentence which will surely leave the reader somewhere in the middle. Yes indeed, a big thank you to the reviewer. |
|
|
Regarding the focus of the review, the author states: “Furthermore, the reviewer appears to have missed the point that the review is not about PDT, but rather about seven specific combinations thereof.” My point was that the focus is never entirely clear at the outset, and this is not unrelated to the issue raised in my previous comment. A clear, simple statement of purpose up front, before the reader goes through many paragraphs of complexly-worded introduction, would be helpful, I think.
|
This point has also finally set. Therefore, I am dedicating an entire paragraph to it, at the end of the introduction, entitled Statement of purpose. I am extracting the material from the various places in the manuscript where it is scattered. Once again, a big thank you to the reviewer. |
|
|
Regarding the overall structure of the manuscript, including clarifying what is novel, what is well-established, etc., I agree with the author that it may not be optimal for certain style preferences, of which there are many. In any case, better use of English and clearer outline of the author’s purpose in writing the review should help.
|
I am grateful that we converge on this point. However, as mentioned above, I am adjusting my style to present clearly the purpose up front. Any writing style where the purpose statement is clarified up front is better. |
|
|
Finally, I do not intend to offend the author and apologize if I have done so. I offer this advice for the sake of the author, the journal, and the readers of the journal. The author is to be commended for compiling and presenting a wide range of information.
|
I have unjustifiably given the reviewer a hard time. I see that now. I am grateful for the generosity of the reviewer and repeat here my sincere apology and appreciation of the work that the reviewer has dedicated to this manuscript. |
|
Comments on the Quality of English Language
|
I have tried to provide the author with some examples of how to possibly make the writing in the manuscript more enlightening and less frustrating for the reader. I personally believe that "moderate" editing is optimal, but I have selected "minor" editing to account for the differences in style noted by the author in the response letter. That is my suggestion, but it's entirely up to the author and Editor(s). |
I would like to acknowledge the reviewer for the recommendation of English editing. As already mentioned, I am appealing to the English editing services of IJMS. |
